# Convergence of Monte Carlo Tree Search in Simultaneous Move Games

**Viliam Lisý**[1]        **Vojtěch Kovařík**[1]        **Marc Lanctot**[2]        **Branislav Bošanský**[1]

[1]Agent Technology Center
Dept. of Computer Science and Engineering
FEE, Czech Technical University in Prague
`<name>.<surname>`
`@agents.fel.cvut.cz`

[2]Department of Knowledge Engineering
Maastricht University, The Netherlands
`marc.lanctot`
`@maastrichtuniversity.nl`

## Abstract

We study Monte Carlo tree search (MCTS) in zero-sum extensive-form games with perfect information and simultaneous moves. We present a general template of MCTS algorithms for these games, which can be instantiated by various selection methods. We formally prove that if a selection method is $\epsilon$-Hannan consistent in a matrix game and satisfies additional requirements on exploration, then the MCTS algorithm eventually converges to an approximate Nash equilibrium (NE) of the extensive-form game. We empirically evaluate this claim using regret matching and Exp3 as the selection methods on randomly generated games and empirically selected worst case games. We confirm the formal result and show that additional MCTS variants also converge to approximate NE on the evaluated games.

## 1 Introduction

Non-cooperative game theory is a formal mathematical framework for describing behavior of interacting self-interested agents. Recent interest has brought significant advancements from the algorithmic perspective and new algorithms have led to many successful applications of game-theoretic models in security domains [1] and to near-optimal play of very large games [2]. We focus on an important class of two-player, zero-sum extensive-form games (EFGs) with perfect information and simultaneous moves. Games in this class capture sequential interactions that can be visualized as a game tree. The nodes correspond to the states of the game, in which both players act simultaneously. We can represent these situations using the normal form (*i.e.,* as matrix games), where the values are computed from the successor sub-games. Many well-known games are instances of this class, including card games such as Goofspiel [3, 4], variants of pursuit-evasion games [5], and several games from general game-playing competition [6].

Simultaneous-move games can be solved exactly in polynomial time using the backward induction algorithm [7, 4], recently improved with alpha-beta pruning [8, 9]. However, the depth-limited search algorithms based on the backward induction require domain knowledge (an evaluation function) and computing the cutoff conditions requires linear programming [8] or using a double-oracle method [9], both of which are computationally expensive. For practical applications and in situations with limited domain knowledge, variants of simulation-based algorithms such as Monte Carlo Tree Search (MCTS) are typically used in practice [10, 11, 12, 13]. In spite of the success of MCTS and namely its variant UCT [14] in practice, there is a lack of theory analyzing MCTS outside two-player perfect-information sequential games. To the best of our knowledge, no convergence guarantees are known for MCTS in games with simultaneous moves or general EFGs.

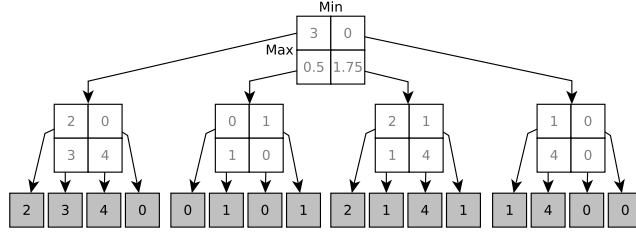

Figure 1: A game tree of a game with perfect information and simultaneous moves. Only the leaves contain the actual rewards; the remaining numbers are the expected reward for the optimal strategy.

In this paper, we present a general template of MCTS algorithms for zero-sum perfect-information simultaneous move games. It can be instantiated using any regret minimizing procedure for matrix games as a function for selecting the next actions to be sampled. We formally prove that if the algorithm uses an $\epsilon$-Hannan consistent selection function, which assures attempting each action infinitely many times, the MCTS algorithm eventually converges to a subgame perfect $\epsilon$-Nash equilibrium of the extensive form game. We empirically evaluate this claim using two different $\epsilon$-Hannan consistent procedures: regret matching [15] and Exp3 [16]. In the experiments on randomly generated and worst case games, we show that the empirical speed of convergence of the algorithms based on our template is comparable to recently proposed MCTS algorithms for these games. We conjecture that many of these algorithms also converge to $\epsilon$-Nash equilibrium and that our formal analysis could be extended to include them.

## 2 Definitions and background

A finite zero-sum game with perfect information and simultaneous moves can be described by a tuple $(\mathcal{N}, \mathcal{H}, \mathcal{Z}, \mathcal{A}, \mathcal{T}, u_1, h_0)$, where $\mathcal{N} = \{1, 2\}$ contains player labels, $\mathcal{H}$ is a set of inner states and $\mathcal{Z}$ denotes the terminal states. $\mathcal{A} = \mathcal{A}_1 \times \mathcal{A}_2$ is the set of joint actions of individual players and we denote $\mathcal{A}_1(h) = \{1 \ldots m^h\}$ and $\mathcal{A}_2(h) = \{1 \ldots n^h\}$ the actions available to individual players in state $h \in \mathcal{H}$. The transition function $\mathcal{T} : \mathcal{H} \times \mathcal{A}_1 \times \mathcal{A}_2 \mapsto \mathcal{H} \cup \mathcal{Z}$ defines the successor state given a current state and actions for both players. For brevity, we sometimes denote $\mathcal{T}(h, i, j) \equiv h_{ij}$. The utility function $u_1 : \mathcal{Z} \mapsto [v_{\min}, v_{\max}] \subseteq \mathbb{R}$ gives the utility of player 1, with $v_{min}$ and $v_{\max}$ denoting the minimum and maximum possible utility respectively. Without loss of generality we assume $v_{\min} = 0$, $v_{\max} = 1$, and $\forall z \in \mathcal{Z}, u_2(z) = 1 - u_1(z)$. The game starts in an initial state $h_0$.

A *matrix game* is a single-stage simultaneous move game with action sets $\mathcal{A}_1$ and $\mathcal{A}_2$. Each entry in the matrix $M = (a_{ij})$ where $(i, j) \in \mathcal{A}_1 \times \mathcal{A}_2$ and $a_{ij} \in [0, 1]$ corresponds to a payoff (to player 1) if row $i$ is chosen by player 1 and column $j$ by player 2. A strategy $\sigma_q \in \Delta(\mathcal{A}_q)$ is a distribution over the actions in $\mathcal{A}_q$. If $\sigma_1$ is represented as a row vector and $\sigma_2$ as a column vector, then the expected value to player 1 when both players play with these strategies is $u_1(\sigma_1, \sigma_2) = \sigma_1 M \sigma_2$. Given a profile $\sigma = (\sigma_1, \sigma_2)$, define the utilities against best response strategies to be $u_1(br, \sigma_2) = \max_{\sigma_1' \in \Delta(\mathcal{A}_1)} \sigma_1' M \sigma_2$ and $u_1(\sigma_1, br) = \min_{\sigma_2' \in \Delta(\mathcal{A}_2)} \sigma_1 M \sigma_2'$. A strategy profile $(\sigma_1, \sigma_2)$ is an $\epsilon$-Nash equilibrium of the matrix game $M$ if and only if

$$u_1(br, \sigma_2) - u_1(\sigma_1, \sigma_2) \leq \epsilon \qquad \text{and} \qquad u_1(\sigma_1, \sigma_2) - u_1(\sigma_1, br) \leq \epsilon \qquad (1)$$

Two-player perfect information games with simultaneous moves are sometimes appropriately called *stacked matrix games* because at every state $h$ each joint action from set $\mathcal{A}_1(h) \times \mathcal{A}_2(h)$ either leads to a terminal state or to a subgame which is itself another stacked matrix game (see Figure 1).

A *behavioral strategy* for player $q$ is a mapping from states $h \in \mathcal{H}$ to a probability distribution over the actions $\mathcal{A}_q(h)$, denoted $\sigma_q(h)$. Given a profile $\sigma = (\sigma_1, \sigma_2)$, define the probability of reaching a terminal state $z$ under $\sigma$ as $\pi^\sigma(z) = \pi_1(z)\pi_2(z)$, where each $\pi_q(z)$ is a product of probabilities of the actions taken by player $q$ along the path to $z$. Define $\Sigma_q$ to be the set of behavioral strategies for player $q$. Then for any strategy profile $\sigma = (\sigma_1, \sigma_2) \in \Sigma_1 \times \Sigma_2$ we define the expected utility of the strategy profile (for player 1) as

$$u(\sigma) = u(\sigma_1, \sigma_2) = \sum_{z \in Z} \pi^\sigma(z) u_1(z) \qquad (2)$$

An $\epsilon$-Nash equilibrium profile $(\sigma_1, \sigma_2)$ in this case is defined analogously to (1). In other words, none of the players can improve their utility by more than $\epsilon$ by deviating unilaterally. If the strategies are an $\epsilon$-NE in each subgame starting in an arbitrary game state, the equilibrium strategy is termed subgame perfect. If $\sigma = (\sigma_1, \sigma_2)$ is an exact Nash equilibrium (*i.e.,* $\epsilon$-NE with $\epsilon = 0$), then we denote the unique value of the game $v^{h_0} = u(\sigma_1, \sigma_2)$. For any $h \in \mathcal{H}$, we denote $v^h$ the value of the subgame rooted in state $h$.

## 3 Simultaneous move Monte-Carlo Tree Search

Monte Carlo Tree Search (MCTS) is a simulation-based state space search algorithm often used in game trees. The nodes in the tree represent game states. The main idea is to iteratively run simulations to a terminal state, incrementally growing a tree rooted at the initial state of the game. In its simplest form, the tree is initially empty and a single leaf is added each iteration. Each simulation starts by visiting nodes in the tree, selecting which actions to take based on a selection function and information maintained in the node. Consequently, it transitions to the successor states. When a node is visited whose immediate children are not all in the tree, the node is expanded by adding a new leaf to the tree. Then, a rollout policy (e.g., random action selection) is applied from the new leaf to a terminal state. The outcome of the simulation is then returned as a reward to the new leaf and the information stored in the tree is updated.

In Simultaneous Move MCTS (SM-MCTS), the main difference is that a joint action of both players is selected. The algorithm has been previously applied, for example in the game of Tron [12], Urban Rivals [11], and in general game-playing [10]. However, guarantees of convergence to NE remain unknown. The convergence to a NE depends critically on the selection and update policies applied, which are even more non-trivial than in purely sequential games. The most popular selection policy in this context (UCB) performs very well in some games [12], but Shafiei et al. [17] show that it does not converge to Nash equilibrium, even in a simple one-stage simultaneous move game. In this paper, we focus on variants of MCTS, which provably converge to (approximate) NE; hence we do not discuss UCB any further. Instead, we describe variants of two other selection algorithms after explaining the abstract SM-MCTS algorithm.

Algorithm 1 describes a single simulation of SM-MCTS. $T$ represents the MCTS tree in which each state is represented by one node. Every node $h$ maintains a cumulative reward sum over all simulations through it, $X_h$, and a visit count $n_h$, both initially set to 0. As depicted in Figure 1, a matrix of references to the children is maintained at each inner node. The critical parts of the algorithm are the updates on lines 8 and 14 and the selection on line 10. Each variant below will describe a different way to select an action and update a node. The standard way of defining the value to send back is $\text{RetVal}(u_1, X_h, n_h) = u_1$, but we discuss also $\text{RetVal}(u_1, X_h, n_h) = X_h/n_h$, which is required for the formal analysis in Section 4. We denote this variant of the algorithms

1  SM-MCTS(node $h$)
2      **if** $h \in \mathcal{Z}$ **then return** $u_1(h)$
3      **else if** $h \in T$ **and** $\exists (i,j) \in \mathcal{A}_1(h) \times \mathcal{A}_2(h)$ *not previously selected* **then**
4          Choose one of the previously unselected $(i,j)$ and $h' \leftarrow \mathcal{T}(h, i, j)$
5          Add $h'$ to $T$
6          $u_1 \leftarrow \text{Rollout}(h')$
7          $X_{h'} \leftarrow X_{h'} + u_1$; $n_{h'} \leftarrow n_{h'} + 1$
8          $\underline{\text{Update}(h, i, j, u_1)}$
9          **return** $\text{RetVal}(u_1, X_{h'}, n_{h'})$
10     $(i,j) \leftarrow \underline{\text{Select}(h)}$
11     $h' \leftarrow \mathcal{T}(h, i, j)$
12     $u_1 \leftarrow \text{SM-MCTS}(h')$
13     $X_h \leftarrow X_h + u_1$; $n_h \leftarrow n_h + 1$
14     $\underline{\text{Update}(h, i, j, u_1)}$
15     **return** $\text{RetVal}(u_1, X_h, n_h)$

**Algorithm 1**: Simultaneous Move Monte Carlo Tree Search

with additional "M" for mean. Algorithm 1 and the variants below are expressed from player 1's perspective. Player 2 does the same except using negated utilities.

## 3.1  Regret matching

This variant applies regret-matching [15] to the current estimated matrix game at each stage. Suppose iterations are numbered from $s \in \{1, 2, 3, \cdots\}$ and at each iteration and each inner node $h$ there is a mixed strategy $\sigma^s(h)$ used by each player, initially set to uniform random: $\sigma^0(h, i) = 1/|\mathcal{A}(h)|$. Each player maintains a cumulative regret $r_h[i]$ for having played $\sigma^s(h)$ instead of $i \in \mathcal{A}_1(h)$. The values are initially set to 0.

On iteration $s$, the Select function (line 10 in Algorithm 1) first builds the player's current strategies from the cumulative regret. Define $x^+ = \max(x, 0)$,

$$\sigma^s(h, a) = \frac{r_h^+[a]}{R_{sum}^+} \text{ if } R_{sum}^+ > 0 \text{ oth. } \frac{1}{|\mathcal{A}_1(h)|}, \text{ where } R_{sum}^+ = \sum_{i \in \mathcal{A}_1(h)} r_h^+[i]. \tag{3}$$

The strategy is computed by assigning higher weight proportionally to actions based on the regret of having not taken them over the long-term. To ensure exploration, an $\gamma$-on-policy sampling procedure is used choosing action $i$ with probability $\gamma/|\mathcal{A}(h)| + (1 - \gamma)\sigma^s(h, i)$, for some $\gamma > 0$.

The Updates on lines 8 and 14 add regret accumulated at the iteration to the regret tables $r_h$. Suppose joint action $(i_1, j_2)$ is sampled from the selection policy and utility $u_1$ is returned from the recursive call on line 12. Define $x(h, i, j) = X_{h_{ij}}$ if $(i, j) \neq (i_1, j_2)$, or $u_1$ otherwise. The updates to the regret are:

$$\forall i' \in \mathcal{A}_1(h), r_h[i'] \leftarrow r_h[i'] + (x(h, i', j) - u_1).$$

## 3.2  Exp3

In Exp3 [16], a player maintains an estimate of the sum of rewards, denoted $x_{h,i}$, and visit counts $n_{h,i}$ for each of their actions $i \in \mathcal{A}_1$. The joint action selected on line 10 is composed of an action independently selected for each player. The probability of sampling action $a$ in Select is

$$\sigma^s(h, a) = \frac{(1 - \gamma) \exp(\eta w_{h,a})}{\sum_{i \in \mathcal{A}_1(h)} \exp(\eta w_{h,i})} + \frac{\gamma}{|\mathcal{A}_1(h)|}, \text{ where } \eta = \frac{\gamma}{|\mathcal{A}_1(h)|} \text{ and } w_{h,i} = x_{h,i}{}^1. \tag{4}$$

The Update after selecting actions $(i, j)$ and obtaining a result $(u_1, u_2)$ updates the visits count $(n_{h,i} \leftarrow n_{h,i} + 1)$ and adds to the corresponding reward sum estimates the reward divided by the probability that the action was played by the player $(x_{h,i} \leftarrow x_{h,i} + u_1/\sigma^s(h, i))$. Dividing the value by the probability of selecting the corresponding action makes $x_{h,i}$ estimate the sum of rewards over all iterations, not only the once where action $i$ was selected.

# 4  Formal analysis

We focus on the eventual convergence to approximate NE, which allows us to make an important simplification: We disregard the incremental building of the tree and assume we have built the complete tree. We show that this will eventually happen with probability 1 and that the statistics collected during the tree building phase cannot prevent the eventual convergence.

The main idea of the proof is to show that the algorithm will eventually converge close to the optimal strategy in the leaf nodes and inductively prove that it will converge also in higher levels of the tree. In order to do that, after introducing the necessary notation, we start by analyzing the situation in simple matrix games, which corresponds mainly to the leaf nodes of the tree. In the inner nodes of the tree, the observed payoffs are imprecise because of the stochastic nature of the selection functions and bias caused by exploration, but the error can be bounded. Hence, we continue with analysis of repeated matrix games with bounded error. Finally, we compose the matrices with bounded errors in

a multi-stage setting to prove convergence guarantees of SM-MCTS. Any proofs that are omitted in the paper are included in the appendix available in the supplementary material and on http://arxiv.org (arXiv:1310.8613).

## 4.1 Notation and definitions

Consider a repeatedly played matrix game where at time $s$ players 1 and 2 choose actions $i_s$ and $j_s$ respectively. We will use the convention $(|\mathcal{A}_1|, |\mathcal{A}_2|) = (m, n)$. Define

$$G(t) = \sum_{s=1}^{t} a_{i_s j_s}, \quad g(t) = \frac{1}{t} G(t), \quad \text{and} \quad G_{max}(t) = \max_{i \in \mathcal{A}_1} \sum_{s=1}^{t} a_{i j_s},$$

where $G(t)$ is the *cumulative payoff*, $g(t)$ is the *average payoff*, and $G_{max}$ is the *maximum cumulative payoff* over all actions, each to player 1 and at time $t$. We also denote $g_{max}(t) = G_{max}(t)/t$ and by $R(t) = G_{max}(t) - G(t)$ and $r(t) = g_{max}(t) - g(t)$ the *cumulative and average regrets*. For actions $i$ of player 1 and $j$ of player 2, we denote $t_i$, $t_j$ the number of times these actions were chosen up to the time $t$ and $t_{ij}$ the number of times both of these actions has been chosen at once. By *empirical frequencies* we mean the strategy profile $(\hat{\sigma}_1(t), \hat{\sigma}_2(t)) \in \langle 0, 1 \rangle^m \times \langle 0, 1 \rangle^n$ given by the formulas $\hat{\sigma}_1(t, i) = t_i/t$, $\hat{\sigma}_2(t, j) = t_j/t$. By *average strategies*, we mean the strategy profile $(\bar{\sigma}_1(t), \bar{\sigma}_2(t))$ given by the formulas $\bar{\sigma}_1(t, i) = \sum_{s=1}^{t} \sigma_1^s(i)/t$, $\bar{\sigma}_2(t, j) = \sum_{s=1}^{t} \sigma_2^s(j)/t$, where $\sigma_1^s$, $\sigma_2^s$ are the strategies used at time $s$.

**Definition 4.1.** We say that a player is $\epsilon$-*Hannan-consistent* if, for any payoff sequences (e.g., against any opponent strategy), $\limsup_{t \to \infty}, r(t) \le \epsilon$ holds almost surely. An algorithm $A$ is $\epsilon$-Hannan consistent, if a player who chooses his actions based on $A$ is $\epsilon$-Hannan consistent.

Hannan consistency (HC) is a commonly studied property in the context of online learning in repeated (single stage) decisions. In particular, RM and variants of Exp3 has been shown to be Hannan consistent in matrix games [15, 16]. In order to ensure that the MCTS algorithm will eventually visit each node infinitely many times, we need the selection function to satisfy the following property.

**Definition 4.2.** We say that $A$ is an *algorithm with guaranteed exploration*, if for players 1 and 2 both using $A$ for action selection $\lim_{t \to \infty} t_{ij} = \infty$ holds almost surely $\forall (i, j) \in \mathcal{A}_1 \times \mathcal{A}_2$.

Note that most of the HC algorithms, namely RM and Exp3, guarantee exploration without any modification. If there is an algorithm without this property, it can be adjusted the following way.

**Definition 4.3.** Let $A$ be an algorithm used for choosing action in a matrix game $M$. For fixed exploration parameter $\gamma \in (0, 1)$ we define a modified algorithm $A^*$ as follows: In each time, with probability $(1 - \gamma)$ run one iteration of $A$ and with probability $\gamma$ choose the action randomly uniformly over available actions, without updating any of the variables belonging to $A$.

## 4.2 Repeated matrix games

First we show that the $\epsilon$-Hannan consistency is not lost due to the additional exploration.

**Lemma 4.4.** *Let $A$ be an $\epsilon$-Hannan consistent algorithm. Then $A^*$ is an $(\epsilon + \gamma)$-Hannan consistent algorithm with guaranteed exploration.*

In previous works on MCTS in our class of games, RM variants generally suggested using the average strategy and Exp3 variants the empirical frequencies to obtain the strategy to be played. The following lemma says there eventually is no difference between the two.

**Lemma 4.5.** *As $t$ approaches infinity, the empirical frequencies and average strategies will almost surely be equal. That is,* $\limsup_{t \to \infty} \max_{i \in \mathcal{A}_1} |\hat{\sigma}_1(t, i) - \bar{\sigma}_1(t, i)| = 0$ *holds with probability* 1.

The proof is a consequence of the martingale version of Strong Law of Large Numbers.

It is well known that two Hannan consistent players will eventually converge to NE (see [18, p. 11] and [19]). We prove a similar result for the approximate versions of the notions.

**Lemma 4.6.** *Let $\epsilon > 0$ be a real number. If both players in a matrix game with value $v$ are $\epsilon$-Hannan consistent, then the following inequalities hold for the empirical frequencies almost surely:*

$$\limsup_{t \to \infty} u\left(br, \hat{\sigma}_2(t)\right) \le v + 2\epsilon \quad \text{and} \quad \liminf_{t \to \infty} u\left(\hat{\sigma}_1(t), br\right) \ge v - 2\epsilon. \tag{5}$$

The proof shows that if the value caused by the empirical frequencies was outside of the interval infinitely many times with positive probability, it would be in contradiction with definition of $\epsilon$-HC. The following corollary is than a direct consequence of this lemma.

**Corollary 4.7.** *If both players in a matrix game are $\epsilon$-Hannan consistent, then there almost surely exists $t_0 \in \mathbb{N}$, such that for every $t \geq t_0$ the empirical frequencies and average strategies form $(4\epsilon + \delta)$-equilibrium for arbitrarly small $\delta > 0$.*

The constant 4 is caused by going from a pair of strategies with best responses within $2\epsilon$ of the game value guaranteed by Lemma 4.6 to the approximate NE, which multiplies the distance by two.

### 4.3  Repeated matrix games with bounded error

After defining the repeated games with error, we present a variant of Lemma 4.6 for these games.

**Definition 4.8.** We define $M(t) = (a_{ij}(t))$ to be a game, in which if players chose actions $i$ and $j$, they receive randomized payoffs $a_{ij}(t, (i_1, ...i_{t-1}), (j_1, ...j_{t-1}))$. We will denote these simply as $a_{ij}(t)$, but in fact they are random variables with values in $[0, 1]$ and their distribution in time $t$ depends on the previous choices of actions. We say that $M(t) = (a_{ij}(t))$ is a *repeated game with error $\eta$*, if there is a matrix game $M = (a_{ij})$ and almost surely exists $t_0 \in \mathbb{N}$, such that $|a_{ij}(t) - a_{ij}| < \eta$ holds for all $t \geq t_0$.

In this context, we will denote $G(t) = \sum_{s \in \{1...t\}} a_{i_s j_s}(s)$ etc. and use tilde for the corresponding variables without errors ($\tilde{G}(t) = \sum a_{i_s j_s}$ etc.). Symbols $v$ and $u(\cdot, \cdot)$ will still be used with respect to $M$ without errors. The following lemma states that even with the errors, $\epsilon$-HC algorithms still converge to an approximate NE of the game.

**Lemma 4.9.** *Let $\epsilon > 0$ and $c \geq 0$. If $M(t)$ is a repeated game with error $c\epsilon$ and both players are $\epsilon$-Hannan consistent then the following inequalities hold almost surely:*

$$\limsup_{t \to \infty} u(br, \hat{\sigma}_2) \leq v + 2(c+1)\epsilon, \quad \liminf_{t \to \infty} u(\hat{\sigma}_1, br) \geq v - 2(c+1)\epsilon \tag{6}$$

$$and \quad v - (c+1)\epsilon \leq \liminf_{t \to \infty} g(t) \leq \limsup_{t \to \infty} g(t) \leq v + (c+1)\epsilon. \tag{7}$$

The proof is similar to the proof of Lemma 4.6. It needs an additional claim that if the algorithm is $\epsilon$-HC with respect to the observed values with errors, it still has a bounded regret with respect to the exact values. In the same way as in the previous subsection, a direct consequence of the lemma is the convergence to an approximate Nash equilibrium.

**Theorem 4.10.** *Let $\epsilon, c > 0$ be real numbers. If $M(t)$ is a repeated game with error $c\epsilon$ and both players are $\epsilon$-Hannan consistent, then for any $\delta > 0$ there almost surely exists $t_0 \in \mathbb{N}$, such that for all $t \geq t_0$ the empirical frequencies form $(4(c+1)\epsilon + \delta)$-equilibrium of the game $M$.*

### 4.4  Perfect-information extensive-form games with simultaneous moves

Now we have all the necessary components to prove the main theorem.

**Theorem 4.11.** *Let $(M^h)_{h \in H}$ be a game with perfect information and simultaneous moves with maximal depth $D$. Then for every $\epsilon$-Hannan consistent algorithm $A$ with guaranteed exploration and arbitrary small $\delta > 0$, there almost surely exists $t_0$, so that the average strategies $(\hat{\sigma}_1(t), \hat{\sigma}_2(t))$ form a subgame perfect*

$$(2D^2 + \delta)\,\epsilon\text{-Nash equilibrium for all } t \geq t_0.$$

Once we have established the convergence of the $\epsilon$-HC algorithms in games with errors, we can proceed by induction. The games in the leaf nodes are simple matrix game so they will eventually converge and they will return the mean reward values in a bounded distance from the actual value of the game (Lemma 4.9 with $c = 0$). As a result, in the level just above the leaf nodes, the $\epsilon$-HC algorithms are playing a matrix game with a bounded error and by Lemma 4.9, they will also eventually return the mean values within a bounded interval. On level $d$ from the leaf nodes, the errors of returned values will be in the order of $d\epsilon$ and players can gain $2d\epsilon$ by deviating. Summing the possible gain of deviations on each level leads to the bound in the theorem. The subgame perfection of the equilibrium results from the fact that for proving the bound on approximation in the whole game (i.e., in the root of the game tree), a smaller bound on approximation of the equilibrium is proven for all subgames in the induction. The formal proof is presented in the appendix.

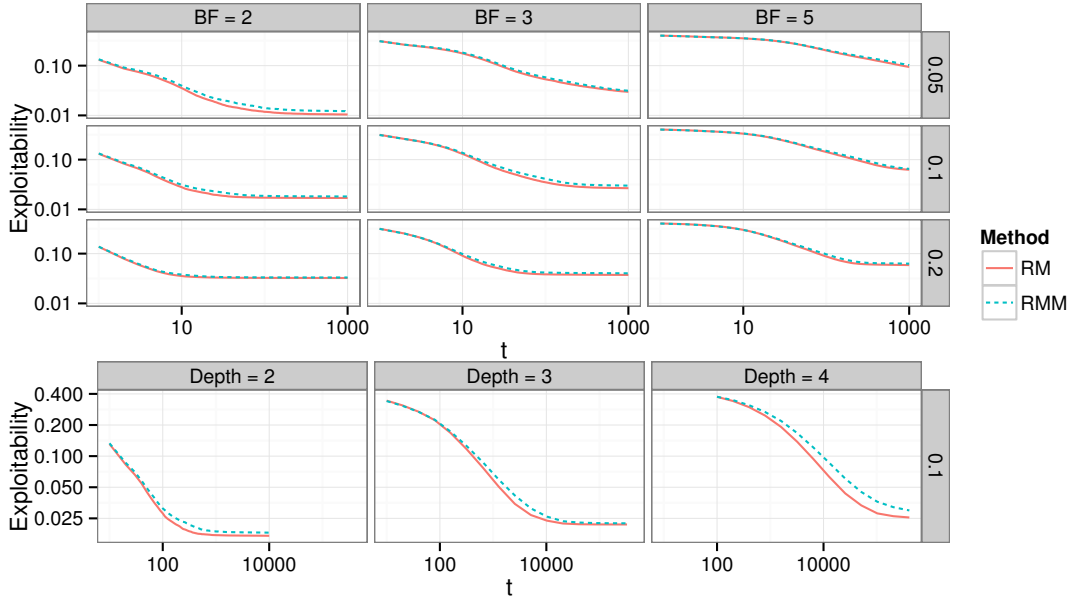

Figure 2: Exploitability of strategies given by the empirical frequencies of Regret matching with propagating values (RM) and means (RMM) for various depths and branching factors.

## 5 Empirical analysis

In this section, we first evaluate the influence of propagating the mean values instead of the current sample value in MCTS to the speed of convergence to Nash equilibrium. Afterwards, we try to assess the convergence rate of the algorithms in the worst case. In most of the experiments, we use as the bases of the SM-MCTS algorithm Regret matching as the selection strategy, because a superior convergence rate bound is known for this algorithm and it has been reported to be very successful also empirically in [20]. We always use the empirical frequencies to create the evaluated strategy and measure the exploitability of the first player's strategy (i.e., $v^{h_0} - u(\hat{\sigma}_1, br)$).

### 5.1 Influence of propagation of the mean

The formal analysis presented in the previous section requires the algorithms to return the mean of all the previous samples instead of the value of the current sample. The latter is generally the case in previous works on SM-MCTS [20, 11]. We run both variants with the Regret matching algorithm on a set of randomly generated games parameterized by depth and branching factor. Branching factor was always the same for both players. For the following experiments, the utility values are randomly selected uniformly from interval $\langle 0, 1 \rangle$. Each experiment uses 100 random games and 100 runs of the algorithm.

Figure 2 presents how the exploitability of the strategies produced by Regret matching with propagation of the mean (RMM) and current sample value (RM) develops with increasing number of iterations. Note that both axes are in logarithmic scale. The top graph is for depth of 2, different branching factors (BF) and $\gamma \in \{0.05, 0.1, 0.2\}$. The bottom one presents different depths for $BF = 2$. The results show that both methods converge to the approximate Nash equilibrium of the game. RMM converges slightly slower in all cases. The difference is very small in small games, but becomes more apparent in games with larger depth.

### 5.2 Empirical convergence rate

Although the formal analysis guarantees the convergence to an $\epsilon$-NE of the game, the rate of the convergence is not given. Therefore, we give an empirical analysis of the convergence and specifically focus on the cases that reached the slowest convergence from a set of evaluated games.

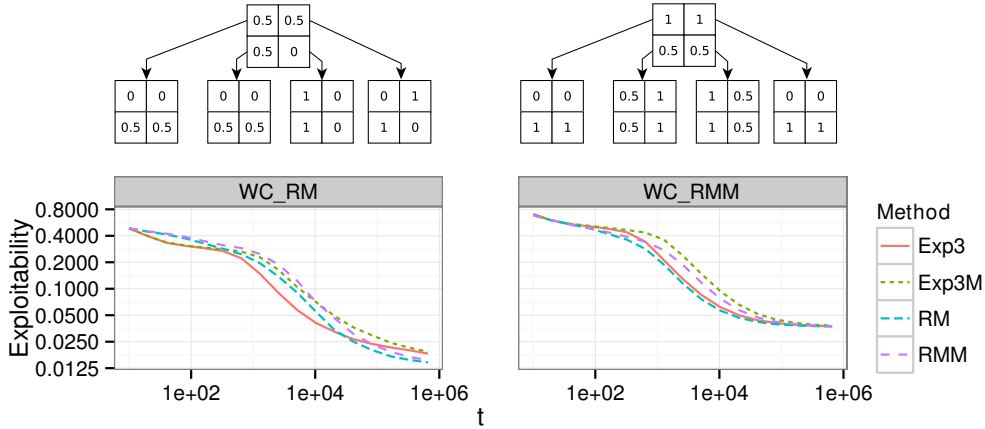

Figure 3: The games with maximal exploitability after 1000 iterations with RM (left) and RMM (right) and the corresponding exploitabililty for all evaluated methods.

We have performed a brute force search through all games of depth 2 with branching factor 2 and utilities form the set $\{0, 0.5, 1\}$. We made 100 runs of RM and RMM with exploration set to $\gamma = 0.05$ for 1000 iterations and computed the mean exploitability of the strategy. The games with the highest exploitability for each method are presented in Figure 3. These games are not guaranteed to be the exact worst case, because of possible error caused by only 100 runs of the algorithm, but they are representatives of particularly difficult cases for the algorithms. In general, the games that are most difficult for one method are difficult also for the other. Note that we systematically searched also for games in which RMM performs better than RM, but this was never the case with sufficient number of runs of the algorithms in the selected games.

Figure 3 shows the convergence of RM and Exp3 with propagating the current sample values and the mean values (RMM and Exp3M) on the empirically worst games for the RM variants. The RM variants converge to the minimal achievable values (0.0119 and 0.0367) after a million iterations. This values corresponds exactly to the exploitability of the optimal strategy combined with the uniform exploration with probability 0.05. The Exp3 variants most likely converge to the same values, however, they did not fully make it in the first million iterations in WC_RM. The convergence rate of all the variants is similar and the variants with propagating means always converge a little slower.

## 6 Conclusion

We present the first formal analysis of convergence of MCTS algorithms in zero-sum extensive-form games with perfect information and simultaneous moves. We show that any $\epsilon$-Hannan consistent algorithm can be used to create a MCTS algorithm that provably converges to an approximate Nash equilibrium of the game. This justifies the usage of the MCTS as an approximation algorithm for this class of games from the perspective of algorithmic game theory. We complement the formal analysis with experimental evaluation that shows that other MCTS variants for this class of games, which are not covered by the proof, also converge to the approximate NE of the game. Hence, we believe that the presented proofs can be generalized to include these cases as well. Besides this, we will focus our future research on providing finite time convergence bounds for these algorithms and generalizing the results to more general classes of extensive-form games with imperfect information.

### Acknowledgments

This work is partially funded by the Czech Science Foundation (grant no. P202/12/2054), the Grant Agency of the Czech Technical University in Prague (grant no. OHK3-060/12), and the Netherlands Organisation for Scientific Research (NWO) in the framework of the project Go4Nature, grant number 612.000.938. The access to computing and storage facilities owned by parties and projects contributing to the National Grid Infrastructure MetaCentrum, provided under the programme "Projects of Large Infrastructure for Research, Development, and Innovations" (LM2010005) is appreciated.

## Footnotes

[1]In practice, we set $w_{h,i} = x_{h,i} - \max_{i' \in \mathcal{A}_1(h)} x_{h,i'}$ since $\exp(x_{h,i})$ can easily cause numerical overflows. This reformulation computes the same values as the original algorithm but is more numerically stable.

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
