[Supplementary Material]

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

## Appendix

*Proof of Lemma 4.4.* Denoting by * the variables corresponding to the algorithm $A^*$ we get

$$r^*(t) = \frac{1}{t} R^*(t) \le \frac{1}{t} \left( 1 \cdot t_{expl} + R(t - t_{expl}) \right) = \frac{t_{expl}}{t} + \frac{R(t - t_{expl})}{t - t_{expl}} \cdot \frac{t - t_{expl}}{t},$$

where, for given $t \in \mathbb{N}$, $t_{expl}$ denotes the number of times $A^*$ explored up to $t$-th iteration. By Strong Law of Large Numbers we have that $\lim\limits_{t \to \infty} \frac{t_{expl}}{t} = \gamma$ holds almost surely. Therefore

$$\limsup_{t \to \infty} r^*(t) \le \limsup_{t \to \infty} \frac{t_{expl}}{t} + \limsup_{t - t_{expl} \to \infty} \frac{R(t - t_{expl})}{t - t_{expl}} \cdot \limsup_{t \to \infty} \frac{t - t_{expl}}{t}$$
$$\le \gamma + \epsilon \left( 1 - \gamma \right)$$
$$\le \gamma + \epsilon,$$

which means that $A^*$ is $(\epsilon + \gamma)$-Hannan consistent. The guaranteed exploration property of $A^*$ is trivial. $\square$

*Remark* 6.1. The guaranteed exploration can be, in theory, achieved even without increasing the approximation factor of Hannan consistency. This method, however, would be impractical in our setting. Fix an increasing sequence of natural numbers $t_n$, such that $\lim\limits_{n \to \infty} \frac{n}{t_n} = 0$ (for example $t_n = 2^n$). Let $A$ be an $\epsilon$-Hannan consistent algorithm. We define modified algorithm $A^+$ as follows: $A^+$ uniformly explores in times $t_n$, $n \in \mathbb{N}$ without modifying the state of $A$ and $A^+$ behaves as $A$ otherwise. Then $A^+$ is an $\epsilon$-Hannan consistent algorithm with guaranteed exploration.

*Proof.* Denoting by + the variables corresponding to the algorithm $A^+$ we get the following inequality for $t \in (t_n, t_{n+1})$:

$$R^+(t) = G^+_{max}(t) - G^+(t) = \max_i \sum_{\substack{s \le t \\ s \ne t_k}} a_{ij_s} - a_{i_s j_s} + \sum_{\substack{s \le t \\ s = t_k}} a_{ij_s} - a_{i_s j_s}$$
$$\le G_{max}(t - n) - G(t - n) + n = R(t - n) + n.$$

Dividing by $t$ and taking $\limsup$ of both sides gives us $\epsilon$-Hannan consistency of $A^+$. Since both players explore at once, and this happens infinitely many times, the guaranteed exploration property of $A^+$ is trivial. $\square$

*Proof of Lemma 4.5.* It is enough to show that $\lim\limits_{t \to \infty} |\hat{\sigma}(t, i) - \bar{\sigma}(t, i)| = 0$ holds almost surely for any given $i$. Using the definitions of $\hat{\sigma}(t, i)$ and $\bar{\sigma}(t, i)$, we get

$$\hat{\sigma}(t, i) - \bar{\sigma}(t, i) = \frac{1}{t} \left( t_i - \sum_{s=1}^{t} \sigma^s(i) \right) = \frac{1}{t} \sum_{s=1}^{t} \left( \delta_{ii_s} - \sigma^s(i) \right),$$

where $\delta_{ij}$ is the Kronecker delta. Using the (martingale difference version of) Strong Law of Large Numbers on the sequence of random variables $X_t = \sum_{s=1}^{t} \left( \delta_{ii_s} - \sigma^s(i) \right)$ gives the result (the conditions clearly hold, since $\mathbf{E} \left[ \delta_{ii_t} - \sigma^t(i) | X_1, ..., X_{t-1} \right] = 0$ implies that $X_t$ is a martingale and $\delta_{ii_t} - \sigma^t(i) \in [-1, 1]$ guarantees that variance is uniformly bounded by 1). $\square$

*Proof of Lemma 4.6.* First we make an observation about $g_{max}$, that will also be useful later on:

$$u\left( br, \hat{\sigma}_2(t) \right) = \max_i \sum_j \hat{\sigma}_2(t, j) a_{ij} = \max_i \sum_j \frac{t_j}{t} a_{ij} = \frac{1}{t} \max_i \sum_j t_j a_{ij}$$
$$= \frac{1}{t} \max_i \sum_{s=1}^{t} a_{ij_s} = \frac{1}{t} G_{max}(t) = g_{max}(t). \tag{8}$$

Now assume for contradiction that with non-zero probability there exists an increasing sequence of time steps $t_n \nearrow \infty$, such that $u\left( br, \hat{\sigma}_2(t_n) \right) \ge v + 2\epsilon + \delta$ for some $\delta > 0$.

Using $\epsilon$-Hannan consistency gives us the existence of such $t_0$, that $g_{max}(t) - g(t) < \epsilon + \frac{\delta}{2}$ holds for all $t \ge t_0$, which is equal to $g(t) > g_{max} - (\epsilon + \frac{\delta}{2})$. However, since we are using a zero-sum matrix game where $g_{max}$ is always at least $v$, this implies that $g(t) > v - (\epsilon + \frac{\delta}{2})$. Using this argument for player 2 gives us that

$$g(t) < v + \epsilon + \frac{\delta}{2}$$

almost surely holds for all $t$ high enough.

This implies that

$$\limsup_{t\to\infty} r(t) \geq \limsup_{n\to\infty} r(t_n) = \limsup_{t\to\infty} g_{max}(t_n) - g(t_n)$$

$$\geq \limsup_{t\to\infty} u\left(br, \hat{\sigma}_2(t_n)\right) - g(t_n) \geq v + 2\epsilon + \delta - (v + \epsilon + \frac{\delta}{2}) > \epsilon$$

holds with non-zero probability, which is in contradiction with $\epsilon$-Hannan consistency. $\square$

*Proof of Corollary 4.7.* This follows immediately from Lemmas 4.5 and 4.6 and the fact that in normal-form zero-sum game with value $v$ the following implication holds:

$$\left(u_1(br, \hat{\sigma}_2) < v + \frac{\epsilon}{2} \quad \text{and} \quad u_1(\hat{\sigma}_1, br) > v - \frac{\epsilon}{2}\right) \Longrightarrow$$

$$(u_1(br, \hat{\sigma}_2) - u_1(\hat{\sigma}_1, \hat{\sigma}_2) < \epsilon \quad \text{and} \quad u_2(\hat{\sigma}_1, br) - u_2(\hat{\sigma}_1, \hat{\sigma}_2) < \epsilon) \overset{\text{def}}{\Longleftrightarrow}$$

$$(\hat{\sigma}_1, \hat{\sigma}_2) \text{ is an } \epsilon\text{-equilibrium.}$$

$\square$

*Remark* 6.2. The following example demonstrates that the extension of the interval in the previous proof is necessary. Consider the following game

| 0.4 | 0.5 |
|-----|-----|
| 0.6 | 0.5 |

and a strategy profile $(1,0),(1,0)$. The value of the game is $v = 0.5$, $u(br,(1,0)) = 0.6$ and $u((1,0),br) = 0.4$. The best responses to the strategies of both players are 0.1 from the game value, but $(1,0),(1,0)$ is a 0.2-NE, since player 2 can improve by 0.2.

*Proof of Lemma 4.9.* This lemma strengthens the result from Lemma 4.6 and its proof will also be similar. The only additional technical ingredient is the following inequality.

Since $M(t)$ is a repeated game with error $c\epsilon$, there almost surely exists $t_0$, such that for all $t \geq t_0$, $|a_{ij}(t) - a_{ij}| \leq c\epsilon$ holds. This leads to:

$$|\tilde{g}_{max}(t) - g_{max}(t)| = \max_i \left| \frac{1}{t} \sum_{s=1}^{t} (a_{ij_s} - a_{ij_s}(s)) \right| \leq \frac{t_0}{t} + c\epsilon \cdot \frac{t - t_0}{t} \overset{t\to\infty}{\longrightarrow} c\epsilon. \tag{9}$$

We are now ready to prove the inequalities in equation (7) from the main paper. Using $\epsilon$-Hannan consistency, we can almost surely bound $g(t)$ for $\delta > 0$ and $t \geq t_0$:

$$g(t) > g_{max}(t) - (\epsilon + \delta)$$
$$\geq \tilde{g}_{max}(t) - (\epsilon + \delta) - |g_{max}(t) - \tilde{g}_{max}(t)|$$
$$\geq v - (\epsilon + \delta) - |g_{max}(t) - \tilde{g}_{max}(t)|.$$

Taking $\liminf$ in the previous inequality and using equation (9) above gives us that

$$\liminf_{t\to\infty} g(t) \geq v - \epsilon - c\epsilon - \delta$$

holds for any $\delta > 0$, therefore $\liminf_{t\to\infty} g(t) \geq v - (c+1)\epsilon$. Applying the same procedure for player 2 will give us the second part of equation (7) from the main paper.

We will prove the left side of inequality (6) from the main paper by contradiction (and omit the proof of the right side since it is identical): assume for contradiction that with non-zero probability there exists an increasing sequence of time steps $t_n \nearrow \infty$, such that

$$u(br, \hat{\sigma}_2(t_n)) \geq v + 2(c+1)\epsilon + \delta \tag{10}$$

holds for some $\delta > 0$. Using equation (8) and equation (9) gives us that there almost surely exists $t_0$, such that for all $t_n \geq t_0$ the following holds:

$$
\begin{aligned}
g_{max}(t_n) &\geq \tilde{g}_{max}(t_n) - |\tilde{g}_{max}(t_n) - g_{max}(t_n)| \\
&= u(br, \hat{\sigma}_2(t_n)) - |\tilde{g}_{max}(t_n) - g_{max}(t_n)| && \text{by equation (8)} \\
&\geq (v + 2(c+1)\epsilon + \delta) - (c\epsilon + \frac{\delta}{2}) && \text{by equations (9), (10)} \\
&= (v + (c+1)\epsilon) + \epsilon + \frac{\delta}{2}
\end{aligned}
$$

and equation (7) from the main paper gives us the inequality

$$
\limsup_{n \to \infty} g(t_n) \leq v + (c+1)\epsilon.
$$

We can now calculate the average regret $r(t_n)$:

$$
\begin{aligned}
\limsup_{n \to \infty} r(t_n) &= \limsup_{n \to \infty} \left( g_{max}(t_n) - g(t_n) \right) \\
&\geq (v + (c+1)\epsilon) + \epsilon + \frac{\delta}{2} - (v + (c+1)\epsilon) \\
&= \epsilon + \frac{\delta}{2}.
\end{aligned}
$$

Therefore $\limsup r(t) > \epsilon$ holds with non-zero probability - a contradiction with the fact that player 1 is $\epsilon$-Hannan consistent. $\qquad\square$

*Proof of Theorem 4.10.* As in the proof of Corollary 4.7, this is an immediate consequence of right-side and left-side inequalities of equation (5) of Lemma 4.6 in the main paper. $\qquad\square$

*Proof of Theorem 4.11.*

*Remark.* In this proof $\delta$ will always be some positive real number. It can vary from term to term, but it can always be made arbitrarily small. We denote the strategies, values, payoffs, etc., in a specific node $h$ by upper index (e.g., $v^h$, $\hat{\sigma}_1^h(t)$, $g^h(t)$).

We prove this using backwards induction. It is enough to verify that the hypothesis of the theorem holds for $C\epsilon-$equilibrium for some constant $C$ and then calculate the value of $C$.

Firstly, it is correct to use all of the above propositions in any $h \in \mathcal{H}$, because $A$ is an algorithm with guaranteed exploration and we will eventually get there infinite number of times. Denote $D$ the depth of the tree, with the root being in depth $d = 1$ and leaves being in depth $d = D$. Then by backwards induction we have:

$d = D$:      If $h \in \mathcal{H}$ is a leaf, then by Lemma 4.6 no player can gain more than $2\epsilon$ utility by deviating from $\left( \hat{\sigma}_1^h(t), \hat{\sigma}_2^h(t) \right)$ for $t$ high enough. By Lemma 4.9, the average payoff $g^h(t)$ will eventually fall into the interval $\left( v^h - \epsilon - \delta, v^h + \epsilon + \delta \right)$.

$(IH_d)$:      Let us denote as induction hypothesis for level $d$ of the game tree the following: In any node $h$ on $\tilde{d}$-th level of the tree, for $d \leq \tilde{d} \leq D$, these two bounds hold:

           $(DB_d)$      No player can gain more than $2(1 + D - \tilde{d})\epsilon + \delta$ by deviating from strategy $\left( \hat{\sigma}_1^h(t), \hat{\sigma}_2^h(t) \right)$.

           $(PB_d)$      The payoff $g^h(t)$ will eventually fall into the interval

$$
\left( v^h - \left(1 + D - \tilde{d}\right)\epsilon - \delta, v^h + \left(1 + D - \tilde{d}\right)\epsilon + \delta \right).
$$

         Since both the deviation bound $(DB_d)$ and the payoff bound $(PB_d)$ hold for $d = D$, we have that $(IH_D)$ holds.

$d \mapsto d - 1$ :      Now we prove the induction step. Assuming that $(IH_d)$ holds. we will prove that $(IH_{d-1})$ holds as well: Fix a node $h \in \mathcal{H}$ in depth $d - 1$ and denote by $h_{ij}$ its children. Note that the nodes $h_{ij}$ are nodes in depth $d$. Then by the induction hypothesis $M(t) = (g^{h_{ij}}(t))_{i,j}$ is a game with error $c\epsilon = (1 + D - d)\epsilon + \delta$. By Lemma 4.9 we have that:

$(A)$         no player can gain more than $2(1 + D - d + 1)\epsilon + \delta = 2(1 + D - (d-1))\epsilon + \delta$ by deviation from $\left(\hat{\sigma}_1^h(t), \hat{\sigma}_2^h(t)\right)$ and

$(B)$         the payoff $g^h(t)$ will eventually fall into the interval

$$\left(v^h - (1 + D - d + 1) - \delta, v^h - (1 + D - d + 1) + \delta\right).$$

Combining $(A)$ for all $h$ on $(d-1)$-th level of the tree with $(DB_d)$ gives $(DB_{d-1})$ and combining $(B)$ for all $h$ on $(d-1)$-th level of the tree with $(PB_d)$ gives $(PB_{d-1})$. Therefore $(IH_{d-1})$ holds and the backwards induction is complete.

Calculation of $C$: Players can gain at most $2\epsilon + \delta$ utility by deviating in leaves, $4\epsilon + \delta$ by deviating one level above the leaves and so on up to the maximum of $2(D-1)\epsilon + \delta$ in the root. Taking the sum of these possible gains, we see that players can possibly gain at most $\frac{D}{2}\left(2 + 2(D-1)\right)\epsilon + D\delta = D^2\epsilon + \delta$ utility. This implies that the pair $(\hat{\sigma}_1(t), \hat{\sigma}_2(t))$ forms $\left(2D^2\epsilon + \delta\right)$-equilibrium for $\delta$ arbitrarily small.

Subgame perfection: We can use the argument with summation of possible improvements over levels for any internal node in the game tree. The sum for any level $d > 1$ will be smaller then the sum in the root; hence, the computed equilibrium is subgame perfect.      $\square$