[Reviews · NeurIPS 2013]

Submitted by Assigned_Reviewer_1

This paper studies Monte Carlo tree search in zero-sum extensive form games with perfect information and simultaneous moves. It is proved that the MCTS algorithm converges to an approximate Nash equilibrium under certain conditions. Empirical study confirms the formal result.

The detailed comments are as follows.
1. Overall I think it is a good paper which provides a sufficient condition for the convergence of MCTS algorithms. The result is useful and the presentation is clear. However, as the main contribution of the paper should be the theoretical part, I have concern on the novelty of the proof technique.
2. UCB algorithms are widely used algorithms in the literature but they are not analyzed in this paper. Given there have been some analysis on the UCB for Trees, it will be better that the authors also consider UCB algorithms and compares them with other algorithms in the paper.
3. It is only proved that the propagation of the mean has good convergence. However, in the experiments the propagation of current sample value also seems good under different criteria. Can you give some explanation?
Summary: This paper studies Monte Carlo tree search in zero-sum extensive form games with perfect information and simultaneous moves. The findings in the paper are interesting, however, technically speaking, the paper is not very outstanding.

Submitted by Assigned_Reviewer_3

This paper analyzes the class of zero-sum perfect-information simultaneous move games. There are some interesting games in this class e.g. goofspiel and Tron. The authors analyze MCTS, implemented with regret-matching for this class of games and prove convergence to an approximate Nash equilibrium.

The paper is clear and well-written, and the results are well-presented. The main contribution is new results in the formal analysis. Previous work, that the authors cited [20] Lanctot et al, 2013, looks at the setup but has more preliminary analysis. The proof is inductive - with optimal strategies at the leaf nodes.

I think the biggest weakness in this type of approach is that not very interesting things can be proved about the inner nodes of the game tree. This misses out on some key ideas in game theory: Subgame perfect nash equilibria. I think the paper can be improved by tying the analysis to those concepts. Recommended citation:
"Existence of subgame perfect equilibria in games with simultaneous moves". C. Harris 1990.
Summary: This paper presents novel analyses of regret-matching strategy with MTCS in zero-sum perfect information simultaneous move games. While interesting from a computational standpoint, the analyses misses out on tying to powerful tools of sub-game perfect Nash equilibria for these class of games; something that has been well-studied by Game Theorists.

Submitted by Assigned_Reviewer_7

The authors present an analysis for a class of Monte Carlo tree search algorithms using essentially (with some mild restrictions), any no-regret selection method. Their primary theoretical contribution is a proof of convergence to epsilon-Nash in the case of simultaneous moves.

One weakness is that the convergence is asymptotic (no rates are provided) which is a necessary consequence of the authors' analysis since the selection method is assumed only to be epsilon-Hannan consistent. This is not necessarily detrimental since, as far as I can tell, there don't seem to exist any proofs of convergence for MCTS methods for extensive-form games with simultaneous moves.

However, I think the question of rates is important, since we know that backwards induction will give us an exact equilibrium anyway. Do the authors know what rates are attained when a specific regret rate is assumed for the selection method, for example, by using the \sqrt{T} guaranteed by exp?

The authors conclude with some experiments. They compare the approach of propagating mean reward values used in their analysis to the standard approach of propagating the current sample value. They find that mean values slightly underperform experimentally. Finally, they attempt to experimentally derive some insight on the convergence rate.

To conclude: This is a well-written paper that is technically correct. I have a mild reservation regarding the ultimate significance of the work given the lack of convergence rates. On the one hand, MCTS methods are knows to work well in practice, so even if rates don't follow from the authors' analysis, establishing asymptotic convergence is a good first step. On the other hand, convergence results are already known for MCTS with sequential moves, so the result feels a little incremental.

Nitpicks
Figure 1: leafs --> leaves
Line 144: negated, not "inverted," right?
Line 10 of Algorithm 1: I think (a_1,a_2) should be (i,j), or possibly (\sigma_1,\sigma_2). This is especially confusing since a_{i,j} was introduced as notation for the payoffs in a single-stage matrix game.
Line 172: It seems like notation hiccups from (i,j) to a's might continue throughout the paper. I'll stop pointing them out, but these should be edited.
Summary: This was a well-written paper, that I enjoyed reading. I have some reservations about its impact, given the lack of convergence rates in the analysis.
Author Feedback

Author rebuttal: We thank all the reviewers for the thoughtful and valuable feedback.

Response to Reviewer 1:

Concerning the novelty of the proof technique: we provide the first convergence proof for MCTS in simultaneous move games. In order for the proof to cover the general class of epsilon-Hannan Consistent selection policies, we define the algorithms with guaranteed exploration and repeated games with bounded error. These notions, their properties, lemmas, and theorems are novel and reusable in other analyses.

We agree that UCB is an important algorithm and it is often used in practice. However, it has been shown not to converge to a Nash equilibrium of a simple (one-state) game [17], as we note in the second paragraph of Section 3. In fact, it can converge to a strategy with large regret, which occurs in practice even in small subgames of Goofspiel [20]. It is for this reason that we chose not to include it in the empirical analysis. However, we agree that it should be discussed and will add more explicit discussion of this point to the paper.

Propagation of values also performs very well in experiments. However, in the proof of Theorem 4.11, if we propagate the current value, the nodes above the leaves will not become a matrix game with epsilon bounded error and the proof is not applicable. A deeper analysis is required and we hope to generalize this in future work.

Response to Reviewer 2:

We agree that subgame perfect equilibria are important in this setting. However, we disagree with the main objection of the reviewer that the biggest weakness in this type of approach is that not very interesting things can be proved about the inner nodes of the game tree. In fact, we can take an intuitive definition of the epsilon-subgame-perfect equilibrium, which would say that playing the computed strategy from any inner node of the game ensures payoff no worse than epsilon far from the actual value of the subgame rooted in the node. With this definition of subgame perfection, our proof shows that the algorithm converges to an epsilon-subgame-perfect equilibrium. The proof is by induction from leaves to the root. In order to prove that the strategy executed from the root is an approximate equilibrium, it proves that executing the strategy from the inner nodes forms an even tighter approximate equilibrium. Furthermore, as reported also by [Harris 90, page 2], there are no issues with existence of the subgame perfect equilibrium as we deal only with finite games. We did not include this discussion as we could not find a definition of epsilon-subgame-perfect equilibrium for this class of games in literature. After this discussion, we admit it was a mistake and that discussion about subgame perfect equilibria and references to relevant literature for this class of games is important. Therefore, we will add it to the final version if accepted for publication.

Response to Reviewer 3:

The question of a finite time regret bound is indeed very interesting and we are currently working on it. However, unless we count in the a positive influence of the random samples executed in MCTS before the tree is completely constructed, at the moment we do not expect to have bounds that would support using this approach rather than full backward induction (in theory). Taking the random samples into account has not been done even in the sequential moves setting and it most likely requires a completely different proof technique. In this paper we focus rather on a clear and intuitive proof, but we are actively investigating a finite time bound for future work.